# Plant diversity effects on forage quality, yield and revenues of semi-natural grasslands

Sergei Schaub [1,2]*, Robert Finger [1], Florian Leiber [3], Stefan Probst [2,4], Michael Kreuzer [2], Alexandra Weigelt [5,6], Nina Buchmann [2] & Michael Scherer-Lorenzen [2,7]

In agricultural settings, plant diversity is often associated with low biomass yield and forage quality, while biodiversity experiments typically find the opposite. We address this controversy by assessing, over 1 year, plant diversity effects on biomass yield, forage quality (i.e. nutritive values), quality-adjusted yield (biomass yield × forage quality), and revenues across different management intensities (extensive to intensive) on subplots of a large-scale grassland biodiversity experiment. Plant diversity substantially increased quality-adjusted yield and revenues. These findings hold for a wide range of management intensities, i.e., fertilization levels and cutting frequencies, in semi-natural grasslands. Plant diversity was an important production factor independent of management intensity, as it enhanced quality-adjusted yield and revenues similarly to increasing fertilization and cutting frequency. Consequently, maintaining and reestablishing plant diversity could be a way to sustainably manage temperate grasslands.

[1] ETH Zürich, Agricultural Economics and Policy Group, Zurich, Switzerland. [2] ETH Zürich, Institute of Agricultural Sciences, Zurich, Switzerland. [3] Research Institute of Organic Agriculture (FiBL), Department of Livestock Sciences, Frick, Switzerland. [4] Bern University of Applied Sciences, School of Agricultural, Forest and Food Sciences, Zollikofen, Switzerland. [5] Leipzig University, Institute of Biology, Leipzig, Germany. [6] German Centre for Integrative Biodiversity Research (iDiv), Halle-Jena-Leipzig, Germany. [7] University of Freiburg, Faculty of Biology, Geobotany, Freiburg, Germany. *email: seschaub@ethz.ch

Grasslands play a crucial role in global food security and are economically important, as they represent an essential basis for milk and meat production in many regions of the world[1,2]. Biomass yield, forage quality (i.e., nutritive values), and the resulting quality-adjusted yield (biomass yield × forage quality) are economically relevant production aspects. Higher plant diversity in agricultural settings is often associated with lower biomass yield and additionally with lower forage quality[3–8], and thus is assumed to have a lower economic value for farmers. This association of higher plant diversity with lower biomass yield and forage quality can be caused by a biased comparison: low-diversity swards in agricultural settings are typically the result of high-intensity management practices, i.e., based on sown, highly productive species, or mixtures (grass-clover) being intensively fertilized, sometimes even on arable land (i.e., intensive or high-input low-diversity systems; sensu Tilman et al.[9]). In contrast, species-diverse (semi-natural) grasslands are often confined to rather unproductive soils and unfavorable climatic conditions. They are typically extensively managed and are nowadays often part of special agri-environmental programs and compensation schemes, which restrict or prohibit fertilization and prescribe late harvests (i.e., extensive or low-input high-diversity systems; sensu Tilman et al.[9]). As a consequence, these diverse swards usually have low annual biomass yield and low forage quality[1,10].

In contrast to these agricultural settings, plant diversity in biodiversity experiments has been shown to increase biomass yield[11–15]. This relationship was also confirmed in experiments along a management intensity gradient, i.e., different fertilization levels and/or cutting frequencies[16–20]. However, findings of a plant diversity effect on forage quality (including contents of crude protein, fiber, energy, and digestibility) in both single and multiple site experiments are ambiguous, and the effects were often reported to be small[21–26]. Important for the productivity of ruminant livestock is the quality-adjusted yield as it represents an integrated measure of biomass yield and forage quality that describes how much quality, for example energy, is available per area. Some studies showed that plant diversity increased quality-adjusted yield, mainly driven by a strong positive effect on biomass yield, also when considering variation in management intensity at a single site[22,25,27–29]. None of these studies considered management intensities from extensive to intensive, a distinct plant diversity gradient from low- to high-diversity systems and a wide range of quality measures at a single site. However, this is required to disentangle the plant diversity effect on biomass yield, forage quality, and quality-adjusted yield from other environmental or management effects, which are always present in agricultural settings.

Furthermore, previous research has shown economic benefits of plant diversity for farmers due to higher revenues and lower production risks when considering biomass yield[30] and forage quality[29,31]. However, there is a lack of evidence on the economic value of plant diversity effects accounting for forage quality and quality-adjusted yields over varying management intensities at a single site, although such assessments could support decision makers in comparing the economic benefits along the increasing plant diversity and management intensity.

To resolve the dichotomy between extensive high-diversity systems and intensive low-diversity systems, we propose a conceptual framework considering both biomass yield and forage quality to assess revenues for milk production (Fig. 1). Here, we focus on four contradicting hypotheses that result from the different observations in agricultural and experimental biodiversity settings:

A. Hypothesis a (Fig. 1a): Biomass yield and forage quality both decrease with increasing plant diversity. Thus, quality-adjusted yield and farm revenues are strongly decreasing with increasing plant diversity.
B. Hypothesis b (Fig. 1b): Biomass yield increases but forage quality decreases with increasing plant diversity at similar

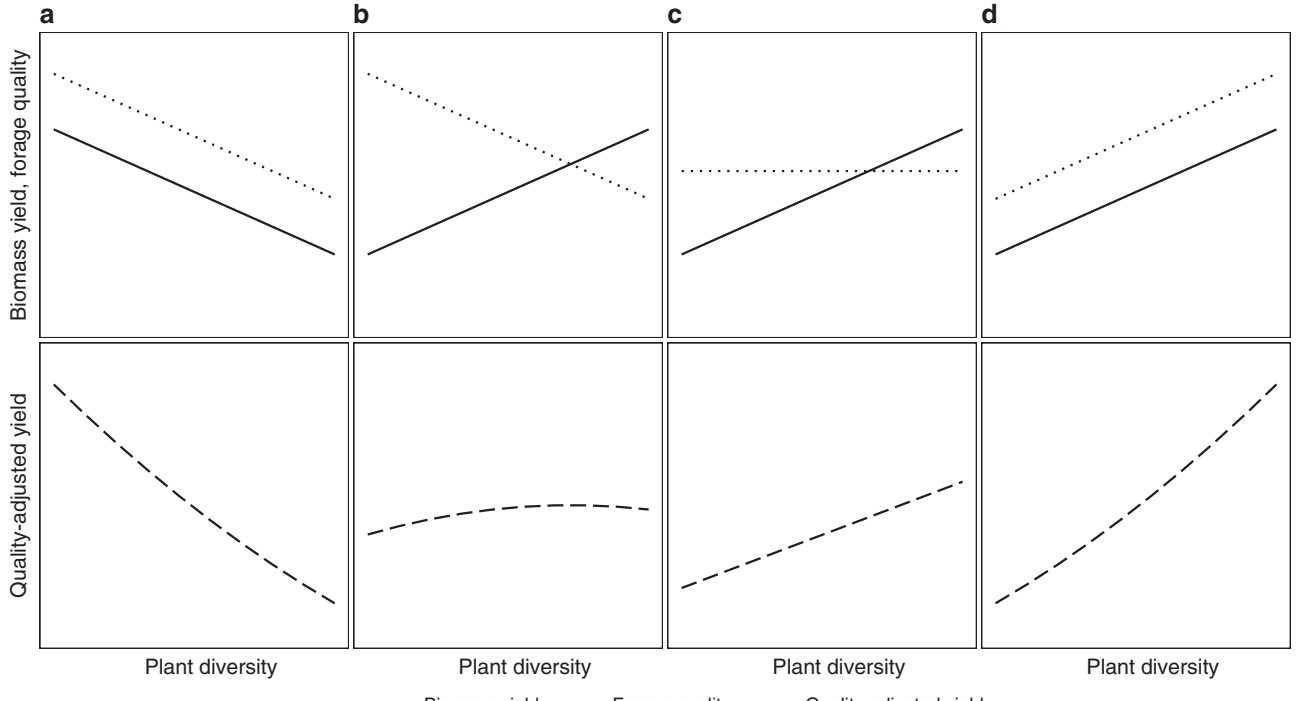

**Fig. 1 Conceptual framework of the relationship between plant diversity and biomass yield, forage quality and quality-adjusted yield.** The panels show the different hypotheses about the plant diversity effect on biomass yield, forage quality, and quality-adjusted yield. **a** Hypothesis a. **b** Hypothesis b. **c** Hypothesis c. **d** Hypothesis d. Source data are provided as a Source Data file.

rates. Hence, quality-adjusted yield and revenues remain constant across different levels of plant diversity.

C. Hypothesis c (Fig. 1c): Biomass yield increases while forage quality stays constant with increasing plant diversity. Hence, quality-adjusted yield and thus revenues increase with increasing plant diversity.

D. Hypothesis d (Fig. 1d): Biomass yield and forage quality both increase with increasing plant diversity. Thus, quality-adjusted yield and farm revenues are strongly increasing with increasing plant diversity.

To test these hypotheses and to compare plant diversity effects with management effects, a biodiversity experiment with different management intensities was set up within a long-term biodiversity experiment (Jena Experiment)[17,32]. This experiment included different plant diversity levels, from 1 to 60 species, and different management intensities, ranging from one cut per year and zero fertilization (extensive) to four cuts and fertilization of 200 kg N ha$^{-1}$ a$^{-1}$ (very highly intensive)[17]. The three intermediate management intensities are defined as: less intensive, intensive, and highly intensive (Supplementary Fig. 1; Supplementary Table 1). The experiment also included different levels of legume shares within each plant diversity level, as nitrogen-fixing legumes play an important role for biomass production and forage quality[33,34]. In our study, we measure relevant forage quality variables (cf. Ball et al.[35]; Supplementary Table 2) and test the effect of plant diversity on biomass yield, different variables of forage quality, quality-adjusted yield, and revenues from potential milk production for different management intensities. We especially focus on metabolizable energy because it is considered a useful measure for overall ruminant-specific nutritional value as it is usually the first limiting factor for ruminant production[10]. We also assess milk production potential and revenues because they represent direct information about animal production and economic implications, which are useful instruments to make better-informed decisions about processes on farm and policy levels.

We find that plant diversity increased quality-adjusted yield and revenues across a wide range of management intensities. Consequently, our findings suggest that maintaining and reestablishing plant diverse grasslands can contribute to sustainable management of temperate grasslands.

## Results

**Forage evaluation.** All measures of quality-adjusted yield (i.e., biomass yield × forage quality), including yields of metabolizable energy (MJ m$^{-2}$ a$^{-1}$), milk production potential (kg m$^{-2}$ a$^{-1}$), crude protein (g m$^{-2}$ a$^{-1}$), utilizable crude protein (g m$^{-2}$ a$^{-1}$), organic matter (g m$^{-2}$ a$^{-1}$), and neutral detergent fiber (g m$^{-2}$ a$^{-1}$) increased significantly with plant diversity, independent of the management intensity (Fig. 2). The only exception was utilizable crude protein yield in the highly intensive management, which was only measured for the first cut of the year. More details about the results, the plant diversity effects on metabolizable energy yield were not significantly different between management intensities (Fig. 3). Increasing the number of species, for example, from 1 to 16 (1–60), the average predicted metabolizable energy yield of all management intensities increased from 4.1 to 6.6 (9.6 MJ m$^{-2}$ a$^{-1}$). The plant diversity effects on milk production potential yield did also not significantly differ between management intensities (Fig. 3). Increasing the number of plant species from 1 to 16 (60) resulted in an average predicted increase of the milk production potential yield of all management intensities from 0.8 to by 1.2 (1.8 kg m$^{-2}$ a$^{-1}$).

Two factors underlie the plant diversity effect on quality-adjusted yield, namely effects on biomass yield and on forage quality. First, we found a positive relationship between plant diversity and biomass yield (g m$^{-2}$ a$^{-1}$), which was robust across all management intensities (Fig. 4). This effect was highest for the intensive management. The difference in the plant diversity effect between this intensive management and the others was significant for all but the extensive management. The plant diversity effects on biomass yield of the other intensities were not significantly different from each other (Fig. 3). Second, the plant diversity effect on forage quality differed among forage quality variables, but the effects were small and insignificant in most of the cases (Fig. 5). More specifically, plant diversity had no effect on metabolizable energy content (MJ kg$^{-1}$ a$^{-1}$) and milk production potential (kg kg$^{-1}$ a$^{-1}$), except a significant and slightly negative effect in the intensive management. When increasing plant diversity from 1 to 16 (60) species, these relative predicted effects for the intensive management on metabolizable energy content and on milk production potential were −4.1% (−9.2%) and −4.9% (−11.1%), respectively. The plant diversity effects on quality-adjusted yield, biomass yield, and forage quality were robust when controlling for legume share instead of legume presence in the analysis (Supplementary Tables 9–14).

Generally, the management effect on quality-adjusted yield and biomass yield increased from extensive to very highly intensive management, except for intensive and highly intensive management which did not differ significantly from each other (Figs. 2–4). The management effect on forage quality was generally as expected, i.e., higher management intensity increased energy and protein contents (Supplementary Tables 3 and 4).

**Economic valuation.** Our results reveal a positive relationship between plant diversity and economic performance, here expressed as revenues from potential milk production (Euro ha$^{-1}$ a$^{-1}$; Fig. 6). This finding was independent of the management intensity. The positive impact of plant diversity on revenues was not significantly different between management intensities (Supplementary Table 8). On average across all management intensities, an increase of plant diversity from 1 to 16 (1–60) plant species increases revenues by about +1400 Euro ha$^{-1}$ a$^{-1}$ (+3100 Euro ha$^{-1}$ a$^{-1}$). Overall, the management effect increased from the extensive to the very highly intensive management, yet with no differences between the intensive and the highly intensive management. It is noteworthy that the management effect of the extensive management was by far the lowest compared to all other management intensities. In economic terms, the predicted change in revenues due to switching from less intensive to intensive was +550 Euro ha$^{-1}$ a$^{-1}$; and +1500 Euro ha$^{-1}$ a$^{-1}$ if switching from less intensive to the very highly intensive management (effect of size a and b respectively in Fig. 6). These effect sizes were of about the same magnitude as changing plant diversity in the less intensive management from 1 to 5 or from 1 to 19 species, respectively.

## Discussion

We presented an analysis of plant diversity effects on different measures of quality-adjusted yield in temperate semi-natural grasslands exposed to different management intensities and we quantified the economic implications of plant diversity. The results show that plant diversity increases quality-adjusted yield by increasing biomass yield at rather constant forage quality. While this does not confirm one individual of our four distinct hypotheses, it supports the notion of hypothesis c (Fig. 1) and nuanced forms of it, i.e., that a strong positive plant diversity effect on biomass yields at rather constant forage quality leads to higher quality-adjusted yield.

The observation of an overall positive and robust influence of plant diversity on quality-adjusted yield, especially based on biomass effects (Supplementary Table 15), is consistent with

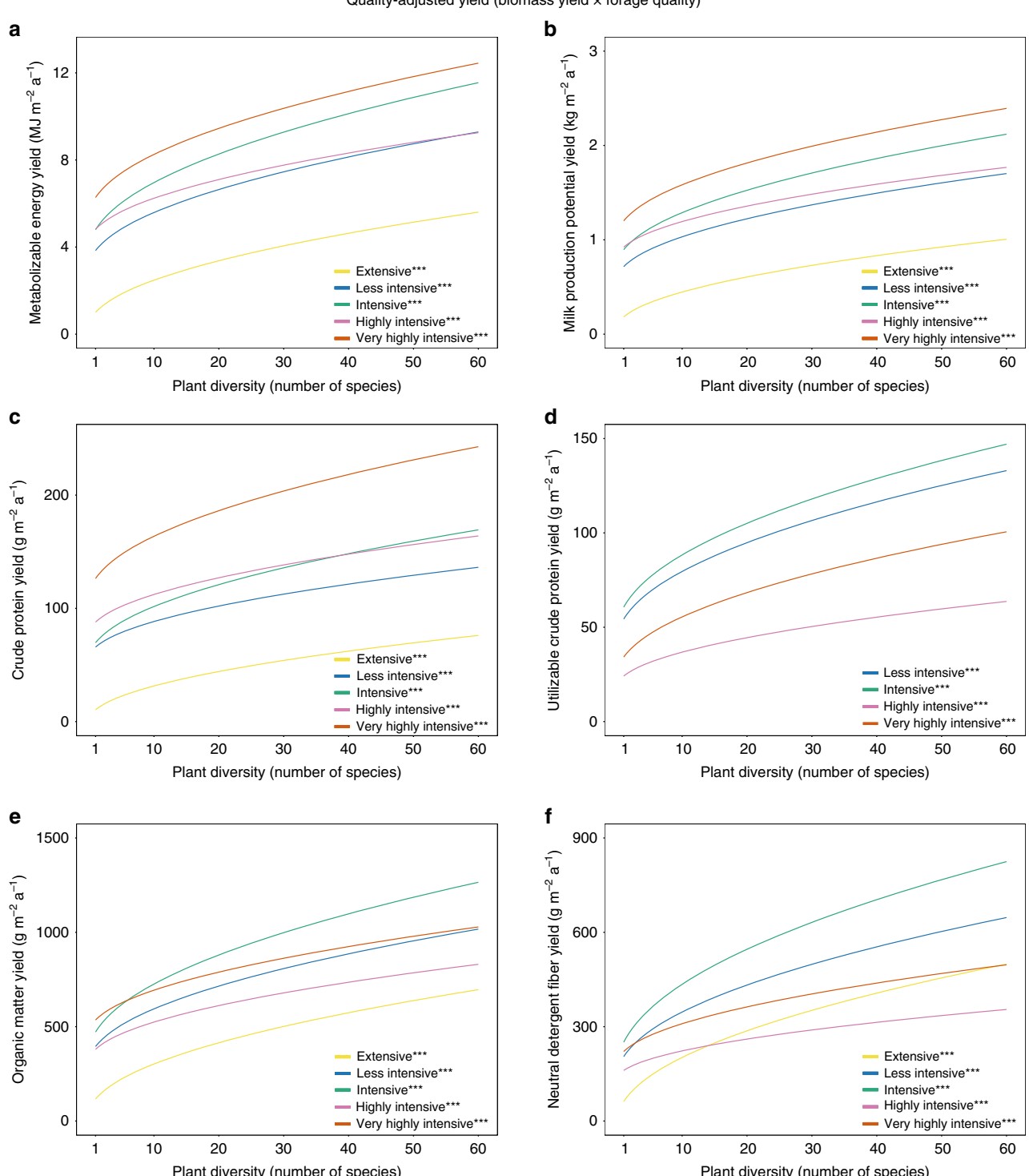

**Fig. 2 Predicted values of annual quality-adjusted yields (i.e., biomass yield × forage quality).** Predicted annual quality-adjusted yields as a function of plant diversity and management intensity include metabolizable energy yield (MJ m$^{-2}$ a$^{-1}$) **a**, milk production potential yield (kg m$^{-2}$ a$^{-1}$) **b**, crude protein yield (g m$^{-2}$ a$^{-1}$) **c**, utilizable crude protein yield (g m$^{-2}$ a$^{-1}$) **d**, organic matter yield (g m$^{-2}$ a$^{-1}$) **e**, and neutral detergent fiber yield (g m$^{-2}$ a$^{-1}$) **f**. *, **, *** denote significance at the 5%, 1%, and 0.1% level of the plant diversity effect per management intensity, respectively (corrected for multiple comparisons). The significance levels are based on a mixed effect model, see Eq. (4). The corresponding coefficients for the plant diversity effect per management intensity can be found in Supplementary Tables 3–5. Utilizable crude protein content was only measured for the first cut of the year. Source data are provided as a Source Data file.

findings from other experiments that investigated yields of crude protein and energy[22,25,27–29]. Further confirmation is given by studies that only reported biomass yield and forage quality but not quality-adjusted yield[21,23,24,26].

The positive plant diversity effect on biomass yield has been related to the complementarity and selection effects[36,37] (but see Barry et. al.[38]). The complementarity effect is assumed to be caused by resource partitioning of different species or positive

| Effect | Variable | Extensive–less intensive | Extensive–intensive | Extensive–highly intensive | Extensive–very highly intensive | Less intensive–intensive | Less intensive–highly intensive | Less intensive–very highly intensive | Intensive–highly intensive | Intensive–very highly intensive | Highly intensive–very highly intensive |
|---|---|---|---|---|---|---|---|---|---|---|---|
| **a** Plant diversity effect per management intensity | Metabolizable energy yield | −0.13 | −0.32 | 0.02 | −0.23 | −0.19 | 0.15 | −0.11 | 0.34 | 0.09 | −0.26 |
| | Milk production potential yield | −0.02 | −0.06 | −0.003 | −0.05 | −0.04 | 0.02 | −0.03 | 0.06 | 0.005 | −0.05 |
| | Crude protein yield | −0.71 | −5.04 | −1.55 | −7.53 | −4.32 | −0.84 | −6.81 | 3.49 | −2.49 | −5.98 |
| | Organic matter yield | −6.28 | −31.73 | 19.15 | 12.85 | −25.45 | 25.43 | 19.13 | 50.88 | 44.58 | −6.3 |
| | Neutral detergent fiber yield | −0.83 | −20.42 | 35.97 | 23.83 | −19.59 | 36.8 | 24.66 | 56.38 | 44.25 | −12.14 |
| | Biomass yield | −7.43 | −35.63 | 18.41 | 10.99 | −28.19 | 25.84 | 18.42 | 54.03 | 46.61 | −7.42 |
| **b** Management effect | Metabolizable energy yield | −1.2 | −3.1 | −3.4 | −5.5 | −1.9 | −2.2 | −4.3 | −0.3 | −2.4 | −2.1 |
| | Milk production potential yield | −0.2 | −0.6 | −0.7 | −1 | −0.4 | −0.4 | −0.8 | −0.1 | −0.5 | −0.4 |
| | Crude protein yield | −19.3 | −38.9 | −65.9 | −109.9 | −19.7 | −46.6 | −90.6 | −27 | −71 | −44 |
| | Organic matter yield | −131.3 | −308.9 | −277.2 | −479.8 | −177.6 | −145.9 | −348.5 | 31.7 | −170.9 | −202.6 |
| | Neutral detergent fiber yield | −71.9 | −176.1 | −138.3 | −226.3 | −104.2 | −66.4 | −154.4 | 37.8 | −50.2 | −88 |
| | Biomass yield | −148.3 | −343.3 | −318.7 | −543.3 | −195 | −170.4 | −395 | 24.6 | −200 | −224.6 |

**Fig. 3 Differences between plant diversity effects per management intensity and management effects.** The figure shows differences and significance of the differences between **a** plant diversity effects per management intensity and **b** management effects on quality-adjusted yield (g or MJ or kg m$^{-2}$ a$^{-1}$) and biomass yield (g m$^{-2}$ a$^{-1}$). Pairs of management intensities indicate the management intensities compared to each other. The displayed numbers are the differences between the effects of the compared management intensities. Light blue, mid blue, and dark blue denote significance at the 5%, 1%, and 0.1% level, respectively (corrected for multiple comparisons), and white denotes no significant effect at the 5% level. The significance levels are based on the Wald test. Note that utilizable crude protein content was only measured for the first cut of the year, which is why we do not compare it between management intensities. The figure is based on Supplementary Tables 3–8.

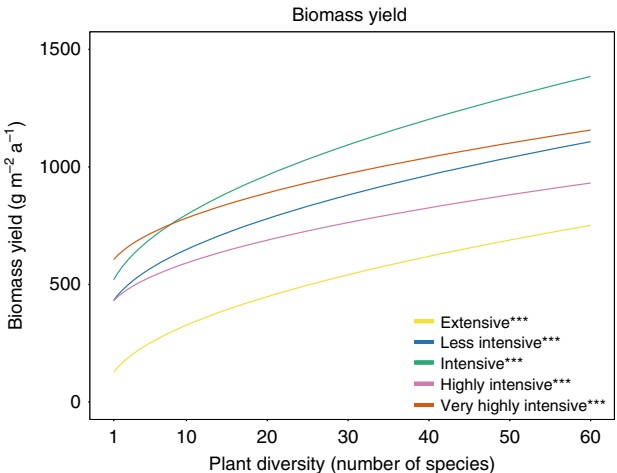

**Fig. 4 Predicted annual biomass yield as a function of plant diversity and management intensity.** *, **, *** denote significance at the 5%, 1%, and 0.1% level of the plant diversity effect per management intensity, respectively. The significance levels are based on a mixed effect model, see Eq. (4). The corresponding coefficients for the plant diversity effect per management intensity can be found in Supplementary Table 6. Source data are provided as a Source Data file.

species interactions, while the selection effect is supposed to be due to a higher probability of including highly productive species in more plant diverse grasslands. For the Jena Experiment, both the complementarity and the selection effects contributed to higher biomass yield with increasing diversity, with the complementarity effect getting stronger and the selection effect getting weaker over time[14,39] (see Weigelt et al.[17] for detailed discussion of biomass yield data).

Furthermore, our analysis demonstrated that strong plant diversity effects on biomass yield in some management intensities were partially counterbalanced by declines in metabolizable energy content and milk production potential. This results in more similar plant diversity effects on metabolizable energy yield and milk production potential yield across all management intensities. This finding is highly relevant when evaluating plant diversity effects on biomass yield across management intensities for agricultural reasons, as it implies that plant diversity might be as important for more intensively managed grasslands as for less intensively managed ones. Ensuring high plant diversity in fertilized grasslands is possible with moderate fertilization, but becomes difficult at very high fertilization levels; however, the species loss in our experiment was slow enough to maintain a distinct plant diversity gradient[17]. Moreover, although many grasslands are permanent, there is a high share of grasslands that are frequently oversown or restored[40–44]. Furthermore, we note that other studies of the same experiment at later years[17,18,27]

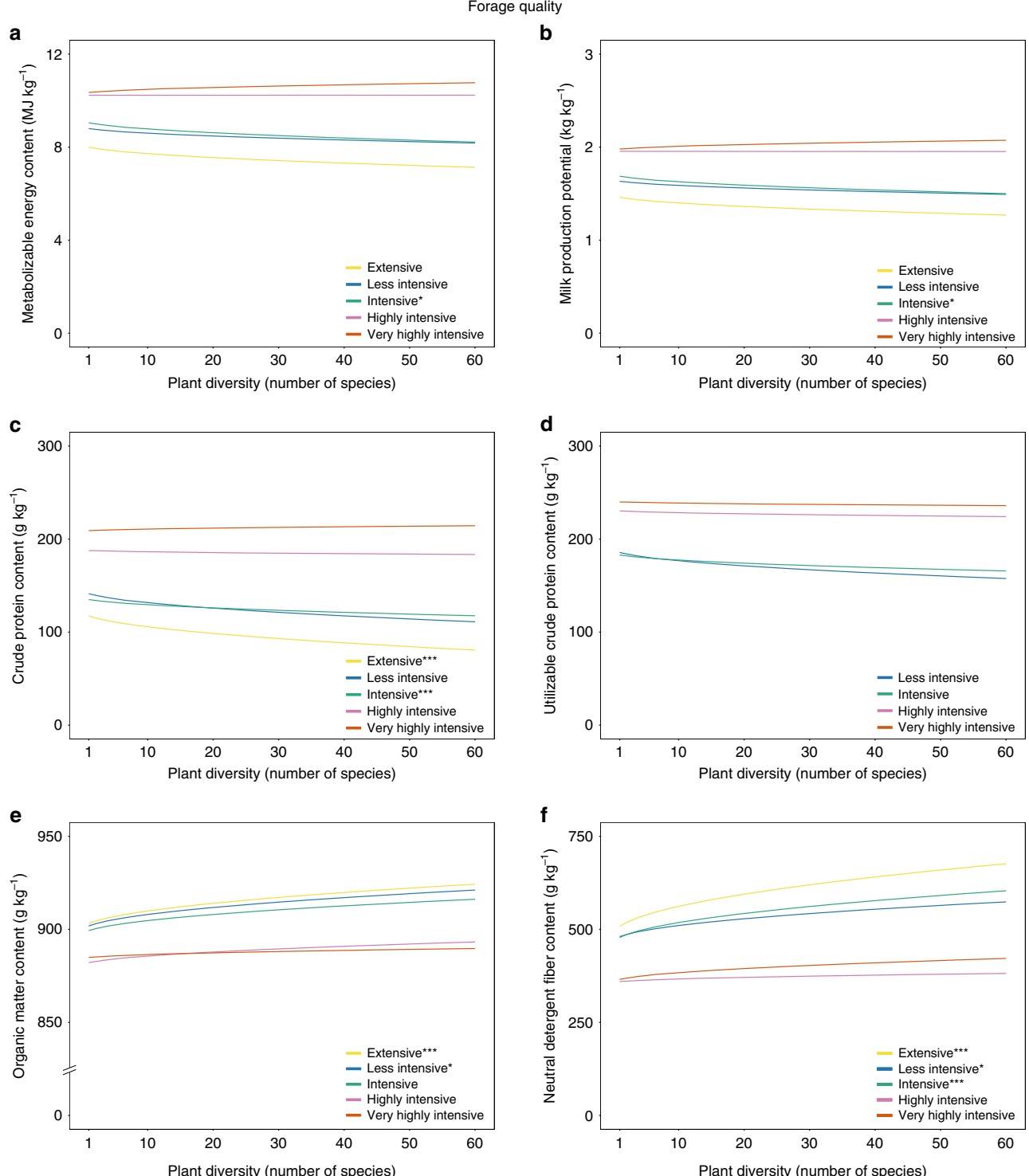

**Fig. 5 Predicted values of annual average forage quality.** Predicted annual average forage quality as a function of plant diversity levels and management intensity include metabolizable energy content (MJ kg$^{-1}$) **a**, milk production potential (kg kg$^{-1}$) **b**, crude protein content (g kg$^{-1}$) **c**, utilizable crude protein content (g kg$^{-1}$) **d**, organic matter content (g kg$^{-1}$) **e**, and neutral detergent fiber content (g kg$^{-1}$;) **f**. *, **, *** denote significance at the 5%, 1%, and 0.1% level of the plant diversity effect per management intensity, respectively (corrected for multiple comparisons). The significance levels are based on a mixed effect model, see Eq. (4). The corresponding coefficients for the plant diversity effect per management intensity can be found in Supplementary Tables 3–5. Utilizable crude protein content was only measured for the first cut of the year. Note that the y-axis break in the figure of organic matter content. Source data are provided as a Source Data file.

support our findings and show robustness of these results also consider longer time horizon.

Furthermore, we also found that plant diversity can achieve gains in increasing metabolizable energy yield and milk production potential yield similar to increasing fertilization levels and/or cutting frequencies. Hence, the production factor plant diversity can reduce other inputs in semi-natural grasslands, while maintaining the same grassland productivity.

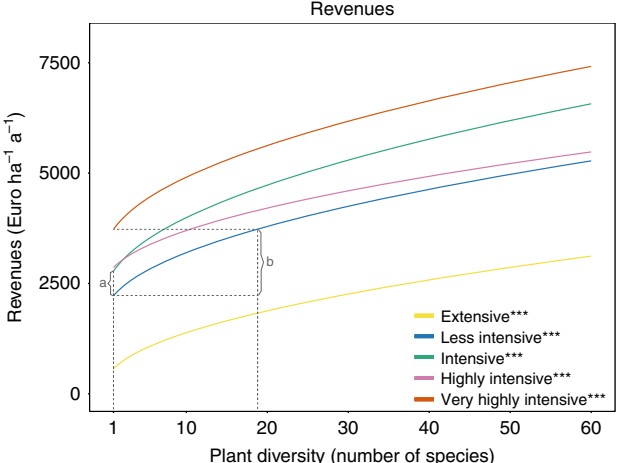

**Fig. 6 Predicted revenues as a function of plant diversity and management intensity.** (**a**) indicates the effect size of changing management intensity from less intensive to intensive, or changing plant diversity in the less intensive management from 1 to 5 species. (**b**) indicates the effect size of changing the management intensity from less intensive to the very highly intensive, or changing plant diversity in the less intensive management from 1 to 19 species.*, **, *** denote significance at the 5%, 1%, and 0.1% level of the plant diversity effect per management intensity, respectively (corrected for multiple comparisons). The significance levels are based on a mixed effect model, see Eq. (4). The corresponding coefficients for the plant diversity effect per management intensity can be found in Supplementary Table 6. Source data are provided as a Source Data file.

Grasslands have important economic, ecological, and cultural functions[1,2,45,46]. The management of these grasslands is becoming more complex, as the traditional interests of production and profit optimization are increasingly complemented by concerns about sustainability and provision of ecosystem services and functions[1,47,48]. Previous studies showed the positive impact of plant diversity on many of these ecosystem services and functions in grasslands[49–54]. In our study we found additionally substantial economic benefits, in terms of revenues, from higher plant diversity. These findings are consistent with previous findings from semi-natural and intensively managed experimental grasslands[29–31]. Our results also show that the plant diversity effect between management intensities becomes more similar when forage quality is included in the economic assessment. This implies that plant diversity in more intensively managed grasslands can be as important as in more extensively managed grasslands. Furthermore, according to our findings, increasing plant diversity in semi-natural grasslands can have equally large positive effects on revenues as increasing management intensity. For example, changing the management intensity from less intensive to intensive (very highly intensive) has the same economic benefit as increasing plant diversity from 1 to 5 (19) plant species in the less intensively managed grassland, namely an effect of ~+550 Euro ha$^{-1}$ a$^{-1}$ or +25% in relative terms (+1500 Euro ha$^{-1}$ a$^{-1}$ or +70% in relative terms) on predicted revenues. However, increasing the management intensity would cause additional variable costs for cutting (including labor and fuel) and fertilization, in our example, i.e., changing management intensity from less intensive to intensive (very highly intensive) management, these costs would be ~−174 (−493) Euro ha$^{-1}$ a$^{-1}$ (see Methods section for calculations). However, we did not consider the costs of seeds of species planted in the Jena Experiment, which can be high for species, which are rarely used in agricultural settings. Similarly, prices for diverse mixtures, which are 'ready to

sow', tend to be expensive. Taking an example of a German seed provider and assuming reseeding of grasslands using available seed mixtures (20 kg ha$^{-1}$), prices of highly diverse mixtures (28–49 species) are considerably higher (mean of 1203 Euro ha$^{-1}$; SD = 521) than that of standard mixtures (1–8 species; mean of 229 Euro ha$^{-1}$; adjusted price; SD = 36). Alternatively, fresh hay transfer, i.e., transferring fresh seed-containing hay from plant diverse grasslands to improve species poor or restore plant diverse grasslands (see e.g., Kiehl et al.[55]), represents a near-natural method and a more cost friendly option, with variable costs of 427 Euro ha$^{-1}$ (including variable fuel, labor, and opportunity costs). However, these costs are depending on the circumstances (see Methods section for details). Therefore, restoring methods aiming at increasing plant diversity can be beneficial for farmers when comparing costs to revenue benefits. Furthermore, less fertilizer application would allow maintaining high plant diversity over longer time[56,57], and in turn maintaining the plant diversity effect. Moreover, any reduction in management can also entail other ecosystem benefits, such as increasing whole-ecosystem biodiversity (beside solely plant diversity) or decreasing greenhouse gas emissions[58,59]. Considering all our findings, altering plant diversity even in more intensively used grasslands appears to be a valuable management tool to farmers.

In conclusion, our findings suggest that increasing plant diversity in semi-natural grasslands presents a viable strategy for sustainable intensification. Plant diversity represents a production factor to increase quality-adjusted yield independent of management intensity. In this respect, plant diversity was found to be as valuable as increasing the fertilization level and cutting frequency in semi-natural grassland. Therefore, we propose that plant diversity should be considered in farm management decisions and in the design of agri-environmental schemes. The challenge remains to develop management systems using mixtures that allow maintaining a high plant diversity also in fertilized grasslands. Here, pathways to exploit the positive plant diversity effects over longer periods could include increasing livestock diversity to promote plant diversity[60], maintaining and promoting species-diverse hay meadows, e.g., *Arrhenatheretum elatioris*, with two to three cuts and low to moderate fertilization levels[61], and seeding of plant diverse mixtures containing complementary species and legumes. Such plant diverse mixtures can also be helpful in dealing with droughts[62–64], which are becoming more severe and frequent under changing climatic conditions[65]. Maintaining and reestablishing plant diverse grasslands could provide a win-win situation as it enables a sustainable increase in quality-adjusted yield and revenues, while at the same time it supports other important ecosystem services and functions.

## Methods

**Experimental design.** Our study is part of the Jena Experiment, a large-scale and long-term biodiversity-ecosystem functioning experiment in Jena (Thuringia, Germany, 50°55' N, 11°35' E, 130 m a.s.l.; mean annual air temperature 9.9 °C, annual precipitation 610 mm; 1980–2010 (ref. [66])).

The experimental communities were established in May 2002, covering different plant diversity levels (including 1, 2, 4, 8, 16, and 60 species) and a functional group gradient (including 1, 2, 3, and 4 functional groups) per plot, and were seeded in 82 main plots of a size of 20 × 20 m, adopting a replacement series design. The species pool consists of 60 species typical to Central European *Arrhenatherum* meadows. Species were categorized into four functional groups, grasses (16 species), small herbs (12), tall herbs (20), and legumes (12) using cluster analysis based on an ecological and morphological trait matrix[32]. The mixtures were assembled by random selection with replacement, yielding 16 replicates for mixtures with 1, 2, 4, and 8 species, and 14 replicates for the 16-species mixtures. In addition, all 60 species were sown on four plots that were used for comparison in the present study. Plots were arranged in four blocks, regularly weeded to maintain the sown plant diversity levels. No fertilization was carried out in the main plots.

The results presented here are from the Management Experiment setup within the Jena Experiment. For the Management Experiment four subplots of 1.6 × 4 m were established in April 2005 within each of the 20 × 20 m main plots. Each

subplot represented one of four additional management intensities with varying fertilization level and cutting frequency per year as listed in Supplementary Table 1 (ref. [17]). The core area of the $20 \times 20$ m main plots served as one management intensity with two cuts per year and zero fertilizer application (less intensive). The five management intensities ranged from extensive to very highly intensive management, including an extensive management (one cut per year, no fertilization) and a very highly intensive management (four cuts per year, high fertilization of 200 kg N ha$^{-1}$ a$^{-1}$ and corresponding P and K fertilization, see below) and three intermediate management intensities: less intensive management (two cuts per year, no fertilization), intensive management (two cuts per year, intermediate fertilization of 100 kg N ha$^{-1}$ a$^{-1}$), and highly intensive management (four cuts per year, intermediate fertilization of 100 kg N ha$^{-1}$ a$^{-1}$). Thus, the experiment consisted of 390 subplots ($82 \times 4$ management subplots plus 82 core areas). We randomized the allocation of management intensities to subplots, except for the extensive subplots, which were always placed at the plot margins due to logistical constraints. The management intensities selected are representative for common grassland management intensities on floodplains comparable to the experimental site, ranging from grasslands in agri-environmental schemes to intensively managed grasslands[17]. We avoided a full factorial design with all fertilization levels per cutting frequency because such a design would include factor combinations that are not reasonable for agricultural practice, such as frequent cutting without fertilization. The controlled manipulation in the experiment of the grassland with different management intensities and different levels of plant diversity, allowed us to test for the presence or absence of a plant diversity effect for different management intensities[67].

For the preparation of the Management Experiment, we fertilized all four subplots dedicated to the experiment once with 50 kg N ha$^{-1}$ a$^{-1}$, 31 kg P$_2$O$_5$ ha$^{-1}$ a$^{-1}$, 31 kg K$_2$O ha$^{-1}$ a$^{-1}$, and 2.75 kg MgO ha$^{-1}$ a$^{-1}$ in April 2005. Starting in 2006, the fertilized subplots received commercial NPK pellets using a lawn fertilizer distributor in amounts presented in Supplementary Table 1. The fertilizer was applied in two equal portions: first in early spring (beginning of April) and second after either the first or second cut (respectively for treatments with two or four cuts) in late June. Plots were cut either once, twice, or four times during the growing season (Supplementary Table 16) with sickle bar mowers at ~3 cm above ground level. All cut material was removed from the plots. Cutting, fertilizing, and weeding were done on a per-block basis such that any maintenance effect was corrected for by the block effect in the statistical analysis.

**Data collection and laboratory analysis**. We measured biomass yield, and several common and relevant forage quality variables in the harvests of 2007 (Supplementary Table 2)[10,35]. Moreover, we estimated contents of metabolizable energy, (metabolically) utilizable crude protein, and milk production potential, all providing valuable information on forage quality related to an agricultural economic perspective.

To measure standing aboveground biomass, we cut all plants within one randomly selected $0.2 \times 0.5$ m frame in each subplot at 3 cm above ground level, shortly before mowing the rest of a subplot. In the main plots, we cut all plants within four randomly selected $0.2 \times 0.5$ m frames at 3 cm above ground level. We oven-dried (70 °C, 48 h) and weighed all harvested biomass of sown species. Subsequently, we milled the samples to pass a 1-mm sieve (rotor mill type SM1, RETSCH, Haan, Germany). Dry matter and total ash content were analyzed in these samples by drying at 105 °C and 550 °C, respectively (AOAC index no. 942.05; AOAC, 1997), with a thermogravimetric determinator furnace (TGA 500, LECO Co., St. Joseph, USA). Organic matter was calculated as dry matter minus total ash. Crude protein content was quantified as $6.25 \times$ nitrogen content using a C/N analyzer (Leco-Analysator Typ FP-2000, Leco Instrumente GmbH, Kirchheim, Germany; AOAC index no. 977.02). Following the procedures of Van Soest et al.[68] using the Fibertec apparatus (Fibertec System M, Tecator, 1020 Hot Extraction, Flawil, Switzerland), we analyzed neutral detergent fiber content, corrected for ash content by addition of sodium sulfite. Contents of ether extract were analyzed with a Soxhlet extractor (Extraktionssystem B-811, Büchi, Flawil, Switzerland; AOAC index no. 963.15). Ether extract content was not reported individually because it is of lower importance as a single variable for forage quality, since it represents only a small share of dry matter, and thus, only of small proportion of energy supply[69,70], but this information was needed for estimating contents of metabolizable and net energy. The content of metabolizable energy was estimated by in vitro fermentation with rumen fluid of a dairy cow, applying the Hohenheim Gas Test procedure[71]. In this approach, 200 mg of feed samples were incubated together with 10 mL of rumen fluid and 20 mL of McDougall buffer for 24 h in glass syringes at 39 °C. Afterward, the gas production was measured by a calibrated scale. Together with compositional information, the content of metabolizable energy was calculated by: metabolizable energy (MJ kg$^{-1}$) = 3.16 + 0.0695 × fermentation gas (mL day$^{-1}$) + 0.000730 × fermentation gas (mL day$^{-1}$)$^2$ + 0.00732 × crude protein (g kg$^{-1}$) + 0.02052 × ether extract (g kg$^{-1}$). Moreover, we estimated the content of net energy for lactation based on the same system, by: net energy (MJ kg$^{-1}$) = 1.64 + 0.0269 × fermentation gas (mL day$^{-1}$) + 0.00078 × fermentation gas (mL day$^{-1}$)$^2$ + 0.0051 × crude protein (g kg$^{-1}$) + 0.01325 × ether extract (g kg$^{-1}$). We subsequently used net energy for lactation to estimate the milk production potential of the biomass yield[72]. In order to estimate utilizable crude protein content, we collected the mixture of rumen fluid and buffer remaining after incubation of the samples in

Falcon tubes and analyzed them for ammonium nitrogen content with the Kjeldahl principle using the distillation unit 323 of Büchi Labortechnik AG (Flawil, Switzerland). Utilizable crude protein content was then calculated by the equation described by Edmunds et al.[73] based on analyzed ammonium content in the incubation fluid, obtained with the Hohenheim Gas test procedure without (blank) and with feed (sample), using an ammonia selective electrode (Metrohm AG, Herisau, Switzerland), and the nitrogen content of the biomass yield samples: utilizable crude protein (g kg$^{-1}$) = [(NH$_3$-N$_{blank}$ + N$_{sample}$ − NH$_3$-N$_{sample}$)/dry matter (mg)] × 6.25 × 1000.

We performed the chemical analyses for the first and the last cut of the year 2007, except for utilizable crude protein content, which we only estimated for the first cut. To retrieve information about the forage quality of harvests from management intensities that included more than two cuts, we used linear interpolation. This was possible, as the forage quality variables showed either a continuous decrease (metabolizable energy content, milk production potential, and crude protein content) or increase (neutral detergent fiber content and ash content) from the first to the last cut. Further, we deleted data from swards that had very small biomass yield or with missing biomass yield information for at least one cut of the year (these were in total 54 swards, from which 31 had a plant diversity level of 1 species, 11 had a plant diversity level of 2 species, 4 had a plant diversity level of 4 species, 7 had a plant diversity level of 8 species, 3 had a plant diversity level of 16 species, and 1 had a plant diversity level of 60 species).

Finally, we calculated the sum of biomass yield of all cuts of a year, i.e., annual biomass yield (g m$^{-2}$ a$^{-1}$), average forage quality of all cuts of a year, i.e., annual average forage quality (g or MJ or kg kg$^{-1}$), and the sum of quality-adjusted yield of all cuts of a year, i.e., annual quality-adjusted yield (g or MJ or kg m$^{-2}$ a$^{-1}$):

$$\text{Biomass yield}_h = \sum_{\text{cut}=1}^{h} \text{Biomass yield}_{\text{cut}} \qquad (1)$$

$$\text{Quality}_h = \sum_{\text{cut}=1}^{h} \text{Quality}_{\text{cut}} \times \frac{\text{Biomass yield}_{\text{cut}}}{\text{Biomass yield}_h} \qquad (2)$$

$$\text{Quality} - \text{adjusted yield}_h = \sum_{\text{cut}=1}^{h} \text{Biomass yield}_{\text{cut}} \times \text{Quality}_{\text{cut}} \qquad (3)$$

$h$ includes all cuts of a year.

**Analysis of the plant diversity effect**. We analyzed the plant diversity effect on biomass yield, forage quality, and quality-adjusted yield using a mixed effect model:

$$y = \alpha + \beta_{D \times M} D^{0.5} xM + \beta_M M + \beta_{L \times M} LxM + \beta_{FG} FG + \beta_G G + \beta_H H + u_B B + u_P P + e \qquad (4)$$

In the Eq. (4), the dependent variable was either annual biomass yield, average annual forage quality or annual quality-adjusted yield. To model the effect of plant diversity, the square root specification ($D^{0.5}$) was chosen over others (linear, linear and squared, logarithmic, and $D^{-1}$), as this specification allowed a diminishing plant diversity effect, which is often observed (see e.g., Hooper et al.[74]), and it performed best across the different outcome variables in terms of the Akaike information criterion and Bayesian information criterion (BIC). More specifically, we modeled the plant diversity effect for each management intensity by introducing an interaction term of the square root of plant diversity and management intensity ($D^{0.5} \times M$). Moreover, we included the different management intensities by a dummy variable for each management intensity ($M$), the interaction term of the presence of legumes with management intensities ($L \times M$), number of functional groups (FG), fixed effects for the presence grasses ($G$), and tall herbs ($H$) as well as random effects for blocks ($B$) and plots ($P$). Finally, we corrected the results for heteroscedasticity by using robust standard errors. In addition, we conducted a robustness analysis, by using a model with the square root of legume share (number of legumes divided by number of all species) instead of the presence of legumes. This allowed us to account in different ways for the importance of legumes[34] and the possibility that the legume share drives the plant diversity effect on nutritive values. Moreover, based on earlier experiments, the Jena Experiment design paid special attention to the role of legumes in grasslands and the interaction with the plant diversity effect, by including legumes in all plant diversity levels, thus avoiding a confounding effect between plant diversity and presence of legumes. However, not every plot included legumes, which offers the possibility to analyze effects of legume presence or abundance[32].

We used the Bonferroni correction to correct for perform multiple comparisons (=significance levels/$n$, $n$ equaled number of different forage qualities, i.e., six, except for biomass yield, for which $n$ was one). Furthermore, we tested whether plant diversity effects per management intensity differed from each other by using a Wald test. To conduct the entire data analyses, we employed Stata 15.0 for Windows.

**Economic valuation**. To evaluate the on-farm value of plant diversity, we computed the annual revenues of milk sales:

$$\text{Revenues}(M, D) = \text{Milk production potential yield}(M, D) \times \text{milk price} \qquad (5)$$

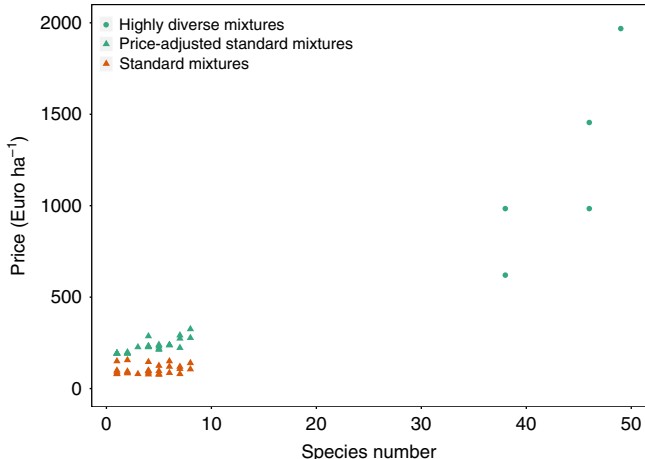

**Fig. 7 Relationship of seed mixtures prices and number of species in the mixture.** The green cycles are highly diverse mixture (38–49 species). Orange triangles are standard mixtures (1–8 species). Green triangles are price-adjusted standard mixtures, based on seed costs of the shop focusing on highly diverse regional mixtures. Source data are provided as a Source Data file.

where milk production potential yield refers to the predicted annual mean milk production potential yield per management intensity for different levels of plant diversity. By using milk revenues, we explicitly considered forage quality. The reference price used for the valuation of the milk production potential yield was the average milk price of 2016 and 2017 in Germany of 0.31 Euro kg$^{-1}$ (ref. [75]). To adapt dimensions of milk production potential yield to more reasonable dimensions from an agricultural economic perspective, we converted the units of milk production potential yield from kg m$^{-2}$ a$^{-1}$ to kg ha$^{-1}$ a$^{-1}$. It shall be emphasized that we assume that farmers maximize their utility, leading to economically efficient decisions (sensu economics). To identify the plant diversity effect, we used the same model as described in Eq. (4) and the Welch $t$-test.

Typical variable fertilizer and cutting costs (including fertilizer, labor, and fuel costs) in Germany were derived from KTBL[76], aiming to represent costs in agricultural settings. Costs of increasing fertilization level of our management intensities by one (Supplementary Table 1) of 165 Euro ha$^{-1}$ a$^{-1}$ were computed by the amount of calcium ammonium nitrate and PK fertilizer required to meet the change in N, P, and K fertilization multiplied with the respective price (100 N ha$^{-1}$/0.27 N kg$^{-1}$ × 0.23 Euro kg$^{-1}$ + max{43.6 P ha$^{-1}$/0.12 P kg$^{-1}$, 83 K ha$^{-1}$/0.24 K kg$^{-1}$} × 0.22 Euro kg$^{-1}$). Additionally, when farmers switch from zero fertilization to some fertilization, variable costs for the process of applying fertilizer (including labor and fuel costs) of 9 Euro ha$^{-1}$ a$^{-1}$ arise (0.55 h ha$^{-1}$ × 13 Euro h$^{-1}$ + 1.9 l ha$^{-1}$ × 0.75 Euro l$^{-1}$). We computed costs of increasing cutting frequency by one cut of 77 Euro ha$^{-1}$ a$^{-1}$ considering labor costs and fuel costs for cutting, windrowing, and collecting the harvest ((0.67 h ha$^{-1}$ + 0.53 h ha$^{-1}$ + 3.56 h ha$^{-1}$) × 13 Euro h$^{-1}$ + (4.85 l ha$^{-1}$ + 3.18 l ha$^{-1}$ + 12.67 l ha$^{-1}$) × 0.75 Euro l$^{-1}$). Furthermore, we included costs of two alternatives for increasing species diversity: reseeding with seed mixtures and fresh hay transfer. The variable costs of reseeding with mixtures comprise two parts, the actual process (reseeding and rolling[76]) and the purchase of the mixture. The process costs (including labor and fuel costs) of 12 Euro ha$^{-1}$ a$^{-1}$ are taken from KTBL[76] for sites 5 km away from the farm ((0.27 h ha$^{-1}$ + 0.41 h ha$^{-1}$) × 13 Euro h$^{-1}$ + (2.08 l ha$^{-1}$ + 2.46 l ha$^{-1}$) × 0.75 Euro l$^{-1}$). For the mixture costs, we collected prices for mixtures from two online retailers (Fig. 7), of which one focuses on highly diverse regional mixtures (highly diverse mixtures; green cycles). We accounted for production cost differences of single seeds between the shops by replicating the mixtures sold at the standard mixture shop (standard mixtures; orange triangles) with seeds of the shop focusing on species-diverse regional mixtures (price-adjusted standard mixtures; green triangles). The species number of the standard and price-adjusted standard mixtures ranges from 1 to 8 and the respective mean (standard deviation) of the prices for reseeding (20 kg ha$^{-1}$) are 104 (26) and 229 (36) Euro ha$^{-1}$ a$^{-1}$. The diverse mixtures include 38–49 species and the mean price (standard deviation) is 1203 (521) Euro ha$^{-1}$ a$^{-1}$.

For deriving reference variable costs for fresh hay transfer, we considered the work steps of cutting, windrowing, collecting, transporting, unloading, and distributing fresh hay[77,78]. Moreover, we assumed the use of machinery, a distance of 5 km between the farm and donating and reseeded grassland sites as well as between sites, a hay transfer ratio of 1:1 (ref. [78]) and no site preparation. We only considered cost for fuel, labor, and opportunity costs, i.e., compensation payment/ forgone revenues. The total variable costs of 427 Euro ha$^{-1}$ a$^{-1}$ ((0.67 h ha$^{-1}$ + 0.53 h ha$^{-1}$ + 5.27 h ha$^{-1}$ + 3.56 h ha$^{-1}$ + 0.89 h ha$^{-1}$) × 13 Euro h$^{-1}$ + (4.85 l ha$^{-1}$ + 3.18 l ha$^{-1}$ + 19.74 l ha$^{-1}$ + 12.67 l ha$^{-1}$ + 5.66 l ha$^{-1}$) × 0.75 Euro l$^{-1}$ + 250 Euro ha$^{-1}$) are derived from KTBL[76] and Kirmer et al.[77]. Note that we always

assumed a distance of 5 km between farm and grassland site, and that all costs can change with equipment used, distances, and other factors.

**Reporting summary**. Further information on research design is available in the Nature Research Reporting Summary linked to this article.

## Data availability

The data used in this study is available at https://www.research-collection.ethz.ch/handle/20.500.11850/374100 [79]. The source data underlying Figs. 1, 2 and 4–7 are provided as a Source Data file. Figure 3 is based on Supplementary Tables 3–8.

## Code availability

The R code (Supplementary Code 1) and Stata code (Supplementary Code 2 and 3) used in this study are available as supplementary material.

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

## Acknowledgements

We thank B. Arnold for helping us to measure forage quality, and all the gardeners and technical staff involved during the experiment. Special thanks goes to D. Bachmann and A. Hämmig for chemical and statistical analyses in earlier versions of this manuscript. The Jena Experiment was financed by the Deutsche Forschungsgemeinschaft (including the project BU1080/4-1) and the Swiss National Science Foundation (FOR 456), and received additional funds from the Friedrich Schiller University Jena, the Max-Planck Institute for Biogeochemistry, Jena and ETH Zurich. This study was further supported by the Mercator Foundation Switzerland within a Zürich-Basel Plant Science Center PhD Fellowship program.

## Author contributions

A.W. and M.S.L. designed the original Management Experiment; A.W., M.S.L., F.L., S.P. and M.K. carried out the biomass and forage quality analyses; S.S., R.F. and N.B. designed, and S.S. and R.F. carried out the economic analyses of the Management Experiment; S.S. re-analysed all data; all authors wrote and commented on this manuscript.

## Competing interests

The authors declare no competing interests.
