## [Peer Review File · Nature Communications]

Reviewers' comments:

Reviewer #1 (Remarks to the Author):

The manuscript by Shaub et al. considers whether and how much plant diversity affects the production and revenues of forage in semi-natural grasslands. The manuscript goes well beyond dozens of previous studies that showed that increasing grassland plant diversity often increases plant biomass production, especially in that it considers forage quality and revenues under a wide range of management intensities. The main results are clear and are succinctly reported: plant diversity substantially increased quality-adjusted yield and revenues. There were surprisingly consistent positive effects of plant diversity on revenues across a range of management scenarios, as nicely shown in Fig. 5. These results are globally relevant because managed grazing is the single largest land use worldwide. The authors are careful not to overstate or extrapolate the results from this single-site study. I think they could go even further, however, toward acknowledging the exceptionally broad relevance of these findings.

My first main suggestion is for the authors to further clarify the strengths and limitations of both their work and that of previous related studies. The Introduction currently seems to suggest that previous studies have focused on the quantity of biomass, but failed to account for the quality of forage and failed to consider a wide range of management scenarios. This summary of the literature misses the contributions of several recent related studies, some of which are already cited, but not given enough credit. There are previous studies that have done a great job of considering effects of biodiversity on both the quantity and the quality of forage production. Some of these previous studies have done an even better job than the present study of considering a wide range of management practices (e.g., Kirwan et al. 2007 *J. Ecol.*, Finn et al. 2013 *J. Appl. Ecol.*, Sturludóttir et al. 2013 *Grass and Forage Science*) or of considering the revenues and costs of diversification (e.g., Binder et al. 2018 *PNAS*). However, none of these previous studies have, to my knowledge, done both: considered effects of biodiversity on revenues of forage production across a wide range of management scenarios. Thus, the greatest novelty of this manuscript by Shaub et al. is that it shows biodiversity can increase revenues under many management scenarios, as very nicely shown in Fig. 5. I do believe this is an important update to this literature. In the Introduction, to better place this novelty in the context of the existing literature, I encourage the authors to give more credit to the many previous studies that have already gone well beyond considering only biodiversity effects on biomass. Rather than using the introduction to say that few studies have done more than look at biomass, it would be much stronger to use the introduction to explain and motivate why it is important to consider revenues, which combine the quantity and quality of biomass, and why it is important to consider a wide range of management strategies, rather than assume results from unmanaged grasslands can be translated to managed grasslands. In other words, given that the Agrodiversity studies have already considered biodiversity effects on the quantity and quality of forage production across a wide range of sites and management practices, what might be gained by considering an integrated measure of quality-adjusted yield or revenues? The answer to this question is not yet a focus of the Introduction, but is central to the novelty and motivation of this manuscript. The hypotheses in Fig. 1 help with this motivation, but do not yet go very far (see more suggestions for improving these hypotheses below). Likewise, given that Binder et al. already

considered quality-adjusted yield and revenues, what might be gained by further considering this across a wide range of management scenarios? For instance, how might adding fertilizer be expected to alter effects of biodiversity on revenues? One possibility that comes to mind is that there may be a ceiling of yield and revenue that can be attained either by fertilizing or by diversifying, but that fertilizing may not be needed at high diversity. This is not what was found, but it is one possible way the introduction could help motivate the need for the novel work presented in this manuscript.

My second main suggestion is to account for the costs of management and diversification. Although it is useful to consider revenues, management decisions are, of course, based not only on revenues, but also on the associated costs. Some effort must be made to account for whether the revenues generated by fertilization or diversification (or conservation of existing diversity) outweigh their added costs. I see that costs of management were mentioned in the Discussion, which is great. It will be important to also include some costs of diversification, even if only in the Discussion, as well. See more specific suggestions on this below.

Additional comments:

I find it confusing that there were both letters, A-D, and numbers 1-4, for the four hypotheses. I recommend that either letters or numbers are chosen and used consistently throughout the manuscript.

Lines 78-89: These four hypotheses fail to include a case that has been previously reported and that I suspect is most often the case: biodiversity strongly increases the quantity of biomass and weakly decreases the quality of biomass, but rather than resulting in no relationship (as in Hypothesis 2), there is instead a positive relationship (as in Hypotheses 3 and 4) because the positive effect of biodiversity on the quantity of biomass production is much larger than the negative effect of biodiversity on the quality of biomass production. At least two of these hypotheses (2 and 3) need to be modified to account for the fact that the relative strengths of the effects on quantity and quality will determine whether the combined effect is positive, neutral, or negative. Indeed, after reading further, I see that your results also show something a bit more nuanced, with mixed and sometimes negative effects on quality components, such that the results do not simply confirm hypothesis C, as is claimed on Line 138.

Line 113: There is a typo here: remove 'from' or 'of'

Line 142: Elaborate a little more on what is meant by 'The management effect on forage quality was as expected...' I suspect the authors mean that fertilization increased crude protein content, but an extra clarifying sentence would help here.

Figure 2: I highly recommend that the authors include a legend that has descriptive words for the management treatments, rather than abbreviations that require more work for the readers.

Figure 2: Rather than add *** in all but one case, I recommend that the authors instead simply note the single non-significant case in the legend.

Figure 4: It was not immediately clear to me what the difference was between Figs 2 and 4. This became clearer after reading the Methods. Given that the Methods are at the end of the manuscript in this journal, I do recommend that the authors make more of an effort in the main text and figure legends to explain why the relationships look so different between these two figures, with nearly identical variables on the axes (though I do see that the units differ).

Figure 5: This figure nicely shows the main novelty of this study. There is, to my knowledge, no other figure like this in the literature. Nicely done!

Line 256: It is very nice to see these costs of changing management included.

Line 257: I do believe you need to report some information on the costs of increasing plant diversity. Although I understand that the costs of some specific species can be expensive, I suspect you could also consider the cost of diverse mixtures of seeds. When such diverse seed mixtures are created by combining many different species that are grown separately, then they are extremely expensive; however, when they are created by harvesting diverse seeds from diverse grasslands, without promising exact proportions of pure live seed by species, then they are much more affordable. Over-seeding with the latter types of seed mixtures might be a relevant estimate for the cost of diversification.

Lines 268-271: The authors might also consider the potential benefits of diversifying livestock, in addition to diversifying plants (e.g., Wang et al. 2019 PNAS, Diversifying livestock promotes multidiversity and multifunctionality in managed grasslands).

Lines 68-72 and 402-404: In both these places, the text implies that Binder et al. (2018 PNAS) did not consider forage quality, which is incorrect. They considered crude protein content and digestibility. The former was included in all analyses and monetary valuation estimates. The latter was excluded from analyses and estimates because there was no evidence that the economic value of forage depended on digestibility. Note that the economic valuation analyses of Shaub et al. implicitly assume perfect economic efficiency, such that anything that actually increases milk production (i.e., the quality measures considered) would be valued by decision makers (e.g., land managers). In reality, however, economic decisions are often sub-optimal, such that decision makers value (and pay more) for things they shouldn't, such as lower-quality forage. Such inefficiencies could be due to imperfect information, negative externalities, or any other sources of market failures. It might be

worthwhile to note that the economic valuation in this manuscript assumes perfectly efficient (sensu economics) markets and decisions.

Finally, I see in the figure legends that tests were corrected for multiple comparisons, but I didn't see in the methods which approach (e.g., Tukey, Bonferoni, etc.) was used.

Reviewer #2 (Remarks to the Author):

This paper reports on an interesting field experiment that addressed plant diversity and management effects on forage yield and nutritive value. Common pool of 60 species found in European grasslands were planted in 5 mixtures of varying diversity levels – including a treatment including all 60 species. The authors found that yield increased with diversity, with nutritive value being largely unaffected across diversity levels. The net effect was an overall positive diversity effect on yield-adjusted forage quality and estimated economic revenues.

At first glance, there was much to like about this paper. Few studies have tried to integrate aspects of forage nutritive value and economic revenues into diversity experiments and relate them to management intensity. Additionally, the overall results were quite intriguing with the suggestion that plant diversity could be as important as management intensity in affecting grassland production and revenues. Despite these positives, I have two significant concerns about this paper.

1) Potential confounding of species composition and diversity effects. As I am sure the authors know, a major determinant of forage nutritive value variation is species composition – mainly due to the presence of legumes. I feel it would have been good to know how much species composition played in the outcome of the diversity relationships. For example, perhaps the main reason nutritive value indices were similar across diversity treatments was because legume composition very similar? Admittedly, separating species composition and diversity effects is difficult and perhaps this was addressed somehow in the statistical analysis. In my mind though, the authors needed to do a better job explaining the potential for this confounding effect given the large impact legumes may have on nutritive value.

2) Study duration. This is a more serious issue. Paper reports on only one year of data (2007) and fertilizers were applied in 2006. This is not enough time to assess the true fertilizer effects on species composition and diversity. There probably is no way diversity levels could be maintained under the high management intensity/N fertilization treatments reported. Presumably, that is probably why the authors are publishing only the 2007 data. So overall, these are interesting results but are probably not reflective of what might happen over longer-term under high N fertilization. The authors need to add several years of data, which presumably they have, to tell a more meaningful

story. Unfortunately, I feel that reporting only one year of data makes the overall take home message of the paper is a bit misleading.

- Reviewer #1 (Remarks to the Author):

The manuscript by Shaub et al. considers whether and how much plant diversity affects the production and revenues of forage in semi-natural grasslands. The manuscript goes well beyond dozens of previous studies that showed that increasing grassland plant diversity often increases plant biomass production, especially in that it considers forage quality and revenues under a wide range of management intensities. The main results are clear and are succinctly reported: plant diversity substantially increased quality-adjusted yield and revenues. There were surprisingly consistent positive effects of plant diversity on revenues across a range of management scenarios, as nicely shown in Fig. 5. These results are globally relevant because managed grazing is the single largest land use worldwide. The authors are careful not to overstate or extrapolate the results from this single-site study. I think they could go even further, however, toward acknowledging the exceptionally broad relevance of these findings.

- (1) My first main suggestion is for the authors to further clarify the strengths and limitations of both their work and that of previous related studies. The Introduction currently seems to suggest that previous studies have focused on the quantity of biomass, but failed to account for the quality of forage and failed to consider a wide range of management scenarios. This summary of the literature misses the contributions of several recent related studies, some of which are already cited, but not given enough credit. There are previous studies that have done a great job of considering effects of biodiversity on both the quantity and the quality of forage production. Some of these previous studies have done an even better job than the present study of considering a wide range of management practices (e.g., Kirwan et al. 2007 J. Ecol., Finn et al. 2013 J. Appl. Ecol., Sturludóttir et al. 2013 Grass and Forage Science) or of considering the revenues and costs of diversification (e.g., Binder et al. 2018 PNAS). However, none of these previous studies have, to my knowledge, done both: considered effects of biodiversity on revenues of forage production across a wide range of management scenarios. Thus, the greatest novelty of this manuscript by Shaub et al. is that it shows biodiversity can increase revenues under many management scenarios, as very nicely shown in Fig. 5. I do believe this is an important update to this literature. In the Introduction, to better place this novelty in the context of the existing literature, I encourage the authors to give more credit to the many previous studies that have already gone well beyond considering only biodiversity effects on biomass. Rather than using the introduction to say that few studies have done more than look at biomass, it would be much stronger to use the introduction to explain and motivate why it is important to consider revenues, which combine the quantity and quality of biomass, and why it is important to consider a wide range of management strategies, rather than assume results from unmanaged grasslands can be translated to managed grasslands. In other words, given that the Agrodiversity studies have already considered biodiversity effects on the quantity and quality of forage production across a wide range of sites and management practices, what might be gained by considering an integrated measure of quality-adjusted yield or revenues? The answer to this question is not yet a focus of the Introduction, but is central to the novelty and motivation of this manuscript. The hypotheses in Fig. 1 help with this motivation, but do not yet go very far (see more suggestions for improving these hypotheses below). Likewise, given that Binder et al. already considered quality-adjusted yield and revenues, what might be gained by further considering this across a wide range of management scenarios? For instance, how might adding fertilizer be expected to alter effects*

of biodiversity on revenues? One possibility that comes to mind is that there may be a ceiling of yield and revenue that can be attained either by fertilizing or by diversifying, but that fertilizing may not be needed at high diversity. This is not what was found, but it is one possible way the introduction could help motivate the need for the novel work presented in this manuscript.

→ Thank you for this input. We re-structured and expanded the introduction to better credit existing work, also including studies on intensively managed grassland. [Line 56 to 63]. We also put more emphasis on the plant diversity effect compared to the management effect. [Line 70 to 73, line to 90 to 92]. We added the references to further studies mentioned by the reviewer. These improvements allow to better position our analysis within the existing research and show the novelty of our approach as pointed out by the reviewer.

- (2) *My second main suggestion is to account for the costs of management and diversification. Although it is useful to consider revenues, management decisions are, of course, based not only on revenues, but also on the associated costs. Some effort must be made to account for whether the revenues generated by fertilization or diversification (or conservation of existing diversity) outweigh their added costs. I see that costs of management were mentioned in the Discussion, which is great. It will be important to also include some costs of diversification, even if only in the Discussion, as well. See more specific suggestions on this below.*

→ We followed the suggestion of the reviewer and added such information in the discussion (see our response to your comment 12 for specifics).

Additional comments:

- (3) *I find it confusing that there were both letters, A-D, and numbers 1-4, for the four hypotheses. I recommend that either letters or numbers are chosen and used consistently throughout the manuscript.*

→ We followed this suggestion. We decided on only using letters. [Fig. 1 and Line 78 to 89].

- (4) *Lines 78-89: These four hypotheses fail to include a case that has been previously reported and that I suspect is most often the case: biodiversity strongly increases the quantity of biomass and weakly decreases the quality of biomass, but rather than resulting in no relationship (as in Hypothesis 2), there is instead a positive relationship (as in Hypotheses 3 and 4) because the positive effect of biodiversity on the quantity of biomass production is much larger than the negative effect of biodiversity on the quality of biomass production. At least two of these hypotheses (2 and 3) need to be modified to account for the fact that the relative strengths of the effects on quantity and quality will determine whether the combined effect is positive, neutral, or negative. Indeed, after reading further, I see that your results also show something a bit more nuanced, with mixed and sometimes negative effects on quality components, such that the results do not simply confirm hypothesis C, as is claimed on Line 138.*

→ Thank you for this remark. We discussed this intensively and decided to keep the four hypotheses and the conceptual representation in Fig. 1. The shift in the slope of the relationships quality with species diversity from negative (Hyp. B), neutral (Hyp. C) to positive (Hyp. D) resulting in no (Hyp. B), slightly (Hyp. C) or strongly positive (Hyp. D) effects on quality-adjusted yields are clearly

distinguishable hypotheses that we can test. The data analysis then shows which of the patterns is present. Adding additional, more nuanced hypotheses would increase the complexity, but also create the next problem where to draw the boundaries for the relative strengths of responses. Instead of increasing the hypothesis from 4 to 7 to 10, we nuanced our text when presenting the data and discussing the four distinctly different hypotheses. This allowed us to keep the number of hypotheses manageable to the reader while following the advice of the reviewer. [Line 214 to 217]

(5) *Line 113: There is a typo here: remove 'from' or 'of'*

→ We corrected the typo. [Line 116].

(6) *Line 142: Elaborate a little more on what is meant by 'The management effect on forage quality was as expected...' I suspect the authors mean that fertilization increased crude protein content, but an extra clarifying sentence would help here.*

→ We agree that this was unclear; we now clarified the statement in the manuscript. [Line 143 to 145].

(7) *Figure 2: I highly recommend that the authors include a legend that has descriptive words for the management treatments, rather than abbreviations that require more work for the readers.*

→ Thank you for this suggestion, we now included the descriptions (extensive, less intensive, intensive, highly intensive and very highly intensive) instead of the abbreviations. [Fig. 2 to 5]. Furthermore, we use now throughout the paper only descriptive words.

(8) *Figure 2: Rather than add *** in all but one case, I recommend that the authors instead simply note the single non-significant case in the legend.*

→ We understand this remark; however, in order to be consistent within the paper (because Fig. 4 has many non-significant results) and with common scientific practice we would like to keep the indication of significances with stars.

(9) *Figure 4: It was not immediately clear to me what the difference was between Figs 2 and 4. This became clearer after reading the Methods. Given that the Methods are at the end of the manuscript in this journal, I do recommend that the authors make more of an effort in the main text and figure legends to explain why the relationships look so different between these two figures, with nearly identical variables on the axes (though I do see that the units differ).*

→ To make this aspect more intuitive, we first added titles and additional descriptions to Fig. 2 and Fig. 4 (as well as to Fig. 3 and Fig. 5 for consistency). Second, to better differentiate between quality-adjusted yield and forage quality when Fig. 2 is first mentioned, we now state again 'quality-adjusted yield (biomass yield × forage quality)'. [Line 113].

(10) *Figure 5: This figure nicely shows the main novelty of this study. There is, to my knowledge, no other figure like this in the literature. Nicely done!*

→ Thank you for these motivating words.

(11) *Line 256: It is very nice to see these costs of changing management included.*

→ Thank you.

(12) *Line 257: I do believe you need to report some information on the costs of increasing plant diversity. Although I understand that the costs of some specific species can be expensive, I suspect you could also consider the cost of diverse mixtures of seeds. When such diverse seed mixtures are created by combining many different species that are grown separately, then they are extremely expensive; however, when they are created by harvesting diverse seeds from diverse grasslands, without promising exact proportions of pure live seed by species, then they are much more affordable. Over-seeding with the latter types of seed mixtures might be a relevant estimate for the cost of diversification.*

→ We followed your suggestions and introduced cost calculations for increasing plant diversity in two ways. First, we considered reseeded with seed mixtures and, second, we included 'fresh hay transfer' as a near-natural restoration method (see e.g. Kiehl et al. 2010). [Line 268 to 275 and line 448 to 475].

(13) *Lines 268-271: The authors might also consider the potential benefits of diversifying livestock, in addition to diversifying plants (e.g., Wang et al. 2019 PNAS, Diversifying livestock promotes multidiversity and multifunctionality in managed grasslands).*

→ Thank you for the input; we included these aspects and cited this paper in the conclusion. [Line 290 to 291].

(14) *Lines 68-72 and 402-404: In both these places, the text implies that Binder et al. (2018 PNAS) did not consider forage quality, which is incorrect. They considered crude protein content and digestibility. The former was included in all analyses and monetary valuation estimates. The latter was excluded from analyses and estimates because there was no evidence that the economic value of forage depended on digestibility. Note that the economic valuation analyses of Shaub et al. implicitly assume perfect economic efficiency, such that anything that actually increases milk production (i.e., the quality measures considered) would be valued by decision makers (e.g., land managers). In reality, however, economic decisions are often sub-optimal, such that decision makers value (and pay more) for things they shouldn't, such as lower-quality forage. Such inefficiencies could be due to imperfect information, negative externalities, or any other sources of market failures. It might be worthwhile to note that the economic valuation in this manuscript assumes perfectly efficient (sensu economics) markets and decisions.*

→ Thank you for this remark. We carefully revised the text and now correctly describe the contribution by Binder et al. (2018). Moreover, we included a clear description of the assumption about efficiency in the discussion as well. [Line 433 to 435].

(15) Finally, I see in the figure legends that tests were corrected for multiple comparisons, but I didn't see in the methods which approach (e.g., Tukey, Bonferoni, etc.) was used.

→ We included the description in the methods. We used the Bonferroni correction to correct for perform multiple comparisons. [Line 421 to 422].

References:

Kiehl, K., Kirmer, A., Donath, T. W., Rasran, L. & Hölzel, N. Species introduction in restoration projects– Evaluation of different techniques for the establishment of semi-natural grasslands in Central and Northwestern Europe. *Basic Appl. Ecol.* **11**, 285-299 (2010).

- Reviewer #2 (Remarks to the Author):

This paper reports on an interesting field experiment that addressed plant diversity and management effects on forage yield and nutritive value. Common pool of 60 species found in European grasslands were planted in 5 mixtures of varying diversity levels – including a treatment including all 60 species. The authors found that yield increased with diversity, with nutritive value being largely unaffected across diversity levels. The net effect was an overall positive diversity effect on yield-adjusted forage quality and estimated economic revenues.

At first glance, there was much to like about this paper. Few studies have tried to integrate aspects of forage nutritive value and economic revenues into diversity experiments and relate them to management intensity. Additionally, the overall results were quite intriguing with the suggestion that plant diversity could as important as management intensity in affecting grassland production and revenues. Despite these positives, I have two significant concerns about this paper.

- i. 1) *Potential confounding of species composition and diversity effects. As I am sure the authors know, a major determinant of forage nutritive value variation is species composition – mainly due to the presence of legumes. I feel it would have been good to know how much species composition played in the outcome of the diversity relationships. For example, perhaps the main reason nutritive value indices were similar across diversity treatments was because legume composition very similar? Admittedly, separating species composition and diversity effects is difficult and perhaps this was addressed somehow in the statistical analysis. In my mind though, the authors needed to do a better job explaining the potential for this confounding effect given the large impact legumes may have on nutritive value.*

→ Thank you for this remark. In the first version of the manuscript, we accounted for legumes in two ways: first, with the design of the Jena Experiment. The design differs from earlier biodiversity experiments by including legumes in all plant diversity levels, thus avoiding a confounding effect between plant diversity and presence of legumes. However, not every plot included legumes, which offers the possibility to analyze effects of legume presence or abundance (Roscher et al. 2004). Second, using the design of the experiment we accounted for legumes in our analysis by controlling for the presence of legumes in a plot.

In the revised manuscript, we now additionally address the influence of legumes in a robustness analysis by accounting for the share of legumes (= number of legume species divided by number of all species) in a plot. The results of this robustness analysis and the main analysis are fairly similar. [Line 136 to 139, line 217, line 412 to 420].

Furthermore, we added information in the main text about the impact and importance of legumes. [Line 96 to 98, 414 to 415]

We also want to emphasize that legumes are indeed often linked to higher crude protein levels. However when legumes are not of high quality because they contain other secondary compounds which are less digestible or when rhizobia do not fix atmospheric N₂ for environmental reasons (e.g. high soil N availability, drought), this is not always true. Under such circumstances, the presence of legumes indeed is not always leading to higher protein and energy contents of the swards (Jayanegara et al. 2011).

- ii. 2) *Study duration. This is a more serious issue. Paper reports on only one year of data (2007) and fertilizers were applied in 2006. This is not enough time to assess the true fertilizer effects on species composition and diversity. There probably is no way diversity levels could be maintained under the high management intensity/N fertilization treatments reported. Presumably, that is probably why the authors are publishing only the 2007 data. So overall, these are interesting results but are probably not reflective of what might happen over longer-term under high N fertilization. The authors need to add several years of data, which presumably they have, to tell a more meaningful story. Unfortunately, I feel that reporting only one year of data makes the overall take home message of the paper is a bit misleading.*

→ We carefully considered your remark, based on which we revised and clarified the message of the manuscript, and we elucidate why the analysis and the (updated) conclusions are valid, even considering the study duration:

First, 2007 was the only year in which the wide range of forage quality measures were assessed within the Management Experiment. Additional measurements of nitrogen content (i.e. linked to crude protein concentrations) within the same Management Experiment are available for 2009, which were collected by Oelmann et al. (2015). They found similar patterns of plant diversity effects as we did in our analysis. This indicates that the plant diversity effect on forage composition remained similar over at least several years. Here we also want to highlight that we used data from the second year (2007, not the first year, 2006) of the experiment, while Oelmann et al. (2015) presented the fourth year of the management experiment. Moreover, fertilization of the plots already started in 2005: fertilized plots had received a pre-fertilization ($50 \text{ kg N ha}^{-1} \text{ year}^{-1}$, $31 \text{ kg P}_2\text{O}_5 \text{ ha}^{-1} \text{ year}^{-1}$, $31 \text{ kg K}_2\text{O ha}^{-1} \text{ year}^{-1}$, and $2.75 \text{ kg MgO ha}^{-1} \text{ year}^{-1}$) already in 2005 (cf. Weigelt et al. 2009). We now refer to this additional information in the revised manuscript. [Line 240 to 241, line 317].

Second, biomass yield was measured within the Management Experiment from 2006 to 2009 (Weigelt et al. 2009, Vogel et al. 2012). The analysis of these data showed that plant diversity increased biomass yield in all years and across all management regimes. Moreover, other research also suggests that the plant diversity effect on biomass yield persists under different fertilization levels and that plant diversity effects on biomass yield increased over time while fertilization effects on biomass yield decreased (Tilman et al. 2012, Craven et al. 2016). Therefore, given that plant diversity does not or only slightly affect forage quality, the results provided here are rather robust. This is an important and novel finding of our study and is contradicting the common assumption that plant diversity generally leads to a lower forage quality. [Line 240 to 241].

Third, although many grasslands are permanent, there is a high share of grasslands that are frequently over-sown or restored; the frequency of grassland renovation and over-sowing varies greatly between regions, management and environmental conditions (Nevens et al. 2002, Schils et al. 2007, Creighton et al. 2011, Lesschen et al. 2014, Eurostat 2019). Thus, the time horizon presented for this experiment in the earlier studies and in this study covers an important time horizon. We added this info now in the discussion to make the relevance of our results clearer. [Line 238 to 240].

We also want to note that the present study does not aim to investigate the effect of management (fertilization level and cutting frequency) on plant diversity, but the plant diversity effect taking place under different management intensities. We thus carefully revised the manuscript to avoid an indication of the first place. Moreover, we revised the introduction in a way to better emphasize this aspect. [Line 109].

Finally, we addressed the reviewer's concerns and revised the conclusion based on the reviewer's remarks. [Line 289 to 294]. More explicitly, we introduced in the conclusion: '... *The challenge remains to develop management systems using mixtures that allow maintaining a high plant diversity with adjusted fertilization schemes over longer times, avoiding frequent reseeding. Here, pathways to exploit the positive plant diversity effects over longer periods could include increasing livestock diversity to promote plant diversity (Wang et al. 2019), maintaining and promoting species diverse hay meadows, e.g. *Arrhenatheretum elatioris*, with two to three cuts and low to moderate fertilization levels (Dierschke and Briemle 2002), and seeding of plant diverse mixtures containing complementary species and legumes...*

References:

- Craven, D., F. et al. Plant diversity effects on grassland productivity are robust to both nutrient enrichment and drought. *Phil. Trans. R. Soc. B* **371**, 20150277 (2016).
- Creighton, P., Kennedy, E., Shalloo, L., Boland, T. M. & O'donovan, M. A survey analysis of grassland dairy farming in Ireland, investigating grassland management, technology adoption and sward renewal. *Grass Forage Sci.* **66**, 251-264 (2011).
- Dierschke, H. & Briemle, G. *Kulturgrasland–Wiesen, Weiden und verwandte Staudenfluren* (Ulmer, Stuttgart, 2002).
- Eurostat Database. Eurostat <https://ec.europa.eu/eurostat/data/database> (2019). Accessed November 2019.
- Finn, J. A. et al. Ecosystem function enhanced by combining four functional types of plant species in intensively managed grassland mixtures: a 3-year continental-scale field experiment. *J. Appl. Ecol.* **50**, 365-375 (2013).
- Jayanegara, A., Marquardt, S., Kreuzer, M., & Leiber, F. Nutrient and energy content, in vitro ruminal fermentation characteristics and methanogenic potential of alpine forage plant species during early summer. *J. Sci. Food Agric.* **91**, 1863-1870 (2011).
- Oelmann, Y., Vogel, A., Wegener, F., Weigelt, A., & Scherer-Lorenzen, M. Management Intensity Modifies Plant Diversity Effects on N Yield and Mineral N in Soil. *Soil Sci. Soc. Am. J.* **79**, 559-568 (2015).
- Lesschen, J. P., Elbersen, B., Hazeu, G. van Doorn, A., Mucher, S. & Velthof G. Task 1 - Defining and classifying grasslands in Europe (Alterra, Wageningen, the Netherlands, 2014).
- Nevens, F., Verbuggen, I., De Vliegheer, A. & Reheul, D. in *Grassland Resowing and Grass-Arable Crop Rotations* Ch. 2 (Plant Research International B.V.,Wageningen, 2002).
- Roscher, C. et al. The role of biodiversity for element cycling and trophic interactions: an experimental approach in a grassland community. *Basic Appl. Ecol.* **5**, 107–121 (2004).
- Schils, R. L. M. et al. in *Grassland Resowing and Grass-Arable Crop Rotations* Ch. 1 (Plant Research International B.V.,Wageningen, 2007).
- Tilman, D., Reich, P. B. & Isbell, F. Biodiversity impacts ecosystem productivity as much as resources, disturbance, or herbivory. *PNAS*, **109**, 10394-10397 (2012).

- Vogel, A., Scherer-Lorenzen, M. & Weigelt, A. Grassland resistance and resilience after drought depends on management intensity and species richness. *PloS ONE*, **7**, e36992 (2012).
- Wang, L. et al. Diversifying livestock promotes multidiversity and multifunctionality in managed grasslands. *PNAS*, **116**, 6187-6192 (2019).
- Weigelt, A., Weisser, W., Buchmann, N. & Scherer-Lorenzen, M. Biodiversity for multifunctional grasslands: equal productivity in high-diversity low-input and low-diversity high-input systems. *Biogeosciences* **6**, 1695-1706 (2009).

REVIEWERS' COMMENTS:

Reviewer #1 (Remarks to the Author):

The authors have sufficiently addressed nearly all of my previous comments and suggestions. In the two cases where they chose not to make the changes I suggested, they offered reasonable counter-points. I appreciate the effort the authors made to revise the manuscript text and display items. The manuscript has substantially improved and was already strong at the time of the original submission.

Reviewer #2 (Remarks to the Author):

The authors have addressed my concerns adequately in the revised MS. I have no other comments.

- Reviewer #1 (Remarks to the Author):

The authors have sufficiently addressed nearly all of my previous comments and suggestions. In the two cases where they chose not to make the changes I suggested, they offered reasonable counterpoints. I appreciate the effort the authors made to revise the manuscript text and display items. The manuscript has substantially improved and was already strong at the time of the original submission.

- Reviewer #2 (Remarks to the Author):

The authors have addressed my concerns adequately in the revised MS. I have no other comments.